# Variations in Pattern of Social Media Engagement between Individuals with Chronic Conditions and Mental Health Conditions

Elizabeth Ayangunna [1], Gulzar Shah [1,*], Kingsley Kalu [1], Padmini Shankar [2] and Bushra Shah [1]

1   Jian-Ping Hsu College of Public Health, Georgia Southern University, Statesboro, GA 30458, USA; ea06435@georgiasouthern.edu (E.A.); kk13870@georgiasouthern.edu (K.K.); bs06779@georgiasouthern.edu (B.S.)
2   Department of Health Sciences & Kinesiology, Georgia Southern University, Statesboro, GA 30458, USA; pshankar@georgiasouthern.edu
*   Correspondence: gshah@georgiasouthern.edu; Tel.: +1-912-478-2419

**Abstract:** The use of the internet and supported apps is at historically unprecedented levels for the exchange of health information. The increasing use of the internet and social media platforms can affect patients' health behavior. This study aims to assess the variations in patterns of social media engagement between individuals diagnosed with either chronic diseases or mental health conditions. Data from four iterations of the Health Information National Trends Survey Cycle 4 from 2017 to 2020 were used for this study with a sample size (N) = 16,092. To analyze the association between the independent variables, reflecting the presence of chronic conditions or mental health conditions, and various levels of social media engagement, descriptive statistics and logistic regression were conducted. Respondents who had at least one chronic condition were more likely to join an internet-based support group (Adjusted Odds Ratio or AOR = 1.5; Confidence Interval, CI = 1.11–1.93) and watch a health-related video on YouTube (AOR = 1.2; CI = 1.01–1.36); respondents with a mental condition were less likely to visit and share health information on social media, join an internet-based support group, and watch a health-related video on YouTube. Race, age, and educational level also influence the choice to watch a health-related video on YouTube. Understanding the pattern of engagement with health-related content on social media and how their online behavior differs based on the patient's medical conditions can lead to the development of more effective and tailored public health interventions that leverage social media platforms.

**Keywords:** social media use; health information exchange; mental health; racial disparities; chronic conditions

## 1. Introduction

Chronic conditions usually involve ongoing treatment and/or limited daily living activities for at least a year or longer [1,2]. Examples of chronic conditions include diabetes, stroke, arthritis, cancer, and heart disease [1,2]. These conditions are the leading causes of mortality and morbidity in the United States as heart disease and stroke cause about 33% of the annual mortality recorded in the nation [3,4]. The prevalence of chronic conditions is alarmingly high in the United States as 60% of adults are recorded to have one chronic condition and 40% have at least two chronic conditions [2,4]. Chronic conditions contribute significantly to healthcare utilization and costs with up to 90% of the country's yearly healthcare expenditure attributed to chronic and mental health conditions [2,4,5]. Preventing and managing chronic conditions is important in improving population health outcomes. Chronic conditions are known to exist together, and there is an increased chance of having multiple chronic conditions as people get older [6].

Mental health conditions often co-exist with chronic conditions due to stress and depression caused by chronic conditions. The financial and activity-limiting burden of chronic conditions can also lead to depression [7]. Mental health conditions refer to medical

conditions that affect individuals' thinking, behavior, feeling, and functioning resulting in a decreased coping capacity for daily activities [8,9]. Mental health conditions such as major depression and anxiety disorders often co-exist together in individuals [8,9]. In the United States, mental illness is one of the most common health conditions. During COVID-19, the prevalence of mental health disorders from meta-analysis was estimated for depression at 31.4%, and for anxiety, distress, and insomnia, it was 31.9%, 41.1%, and 37.9%, respectively [10]. In the U.S., 52.9 million adults aged 18 years and older are living with mental illness, with the highest prevalence reported among White adults (23%) and the lowest prevalence among Asian adults (14%) [11,12]. The number of females suffering from mental health disorders in the U.S. is about 29 million, which is almost twice the number of males (15 million) suffering from these disorders. Mental and behavioral disorders contribute significantly to total health expenditures in the United States, estimated at USD 107 billion in 2019 and USD 238 billion in 2020 [13,14].

The advent of the internet led to the development of social media which patients use for various health-related activities. Patients use social media to seek health information and communicate with providers and other individuals with similar health conditions who can also serve as a source of support [15–17]. Healthy and sick individuals alike seek and use health information on the internet/social media for personal reasons. Those reasons include as self-care, personal health assessment, and to compare healthcare providers and be better caregivers of their loved ones [18,19].

There is an increasing focus on patient-centric care, and about 50% of the public feels that patients themselves should be responsible for preventing chronic diseases [20]. Consequently, the population is increasingly aware of health conditions, leading to patients participating more actively in their personal or loved ones' care. With the increase in the number of patients going online to obtain health information, there have been some concerns about the quality of health information accessed online [18]. Patients now have a wide array of resources to obtain information unlike in the past when healthcare providers were solely responsible for that purpose. Healthcare providers now find themselves answering more questions from patients who have been exposed to a lot of information through social media [21]. Although the use of online health information empowers patients and can help them find information that best relates to their individual needs, which helps them in medical decision-making, such information may also lead to a decline in patients' trust in doctors and the quality of the relationship between doctors and their patients [21].

With access to the internet, individuals routinely seek and find information about diseases, prevention, diagnoses, treatments, and how they affect their quality of life [22–24]. The internet has also allowed a two-way communication channel between providers and patients. This health information is usually accessible through websites, blogs, online health communities, and video channels [15,18]. Generally, health information-seeking behavior was strongly correlated with the female gender, education, gender, age, and socioeconomic status [25,26].

Several researchers have explored the different ways in which patients have used social media for health-related reasons regardless of health status. However, for this study, chronic conditions are treated separately from mental health conditions due to the difference in pathology with the former affecting physical health and the latter affecting the mind. The current study is unique because it examines how the use of social media for health-related reasons differs between patients with chronic conditions, specifically diabetes, hypertension, heart condition, and chronic lung disease, versus mental health conditions, i.e., anxiety and depression. This study specifically aims to assess the differences in the patterns of social media engagement between individuals with chronic conditions and mental health conditions by visiting social media, sharing health information on social media, joining a support group for people with similar conditions, and watching a health-related video on YouTube.

## 2. Materials and Methods

### 2.1. Data Source

This study used the data from the Health Information National Trends Survey (HINTS) 5 after pooling its four cycles (2017–2020). The National Cancer Institute conducts this survey using a cross-sectional study design to collect nationally representative data on the knowledge, attitudes, and use of health information technology among non-institutionalized residents of the United States, 18 years and older. HINTS collected data from a total of 16,092 adults participating in the four cycles, and dataset owners ensured responses were not duplicated from each household. A two-stage sampling design was used in which a stratified sample of addresses was selected from a database of addresses and then an adult within each household was selected to complete the survey. The survey was collected by mail, and respondents were given a $2 (U.S.) incentive for participation. More details about the survey can be found at https://hints.cancer.gov/data/methodology-reports.aspx (accessed on 12 January 2024). The first cycle in 2017 had 3285 respondents, 2018 had 3504 respondents, 2019 had 5438 respondents, and 2020 had 3865 respondents, with more than 30% response rates across the cycles. Georgia Southern University Institutional Review Board approved this study under study protocol H22399.

### 2.2. Variables

#### 2.2.1. Dependent Variables

The dependent variables of interest measuring social media engagement were whether or not the study participants demonstrated any of the following: (a) visited social media, (b) shared health information on social media, (c) joined a support group for people with similar conditions, and (d) watched a health-related video on YouTube. The survey question, "In the last 12 months, have you used the Internet to visit a social networking site, such as Facebook or LinkedIn?" measured the variable "visited social media". The question "In the last 12 months, have you used the Internet to share health information on social networking sites, such as Facebook or Twitter?" measured the variable "shared health information on social media". The variable "joined a support group for people with similar conditions" was operationalized through the survey question "In the last 12 months, have you used the Internet to participate in an online forum or support group for people with a similar health or medical issue?" The survey question "In the last 12 months, have you used the Internet to watch a health-related video on YouTube?" operationalized the fourth dependent variable "watched a health-related video on YouTube". All four dependent variables were dichotomous and coded as yes or no.

#### 2.2.2. Independent Variables

Our inferential analysis consisted of two independent variables of primary interest, the presence of chronic conditions (yes/no) and the presence of mental health (yes/no). The variable presence of "chronic conditions" was based on answering yes to any of the survey questions that asked, "Has a doctor or other health professional ever told you that you had any of the following medical conditions: diabetes or high blood sugar; high blood pressure or hypertension; a heart condition such as heart attack, angina, or congestive heart failure; or chronic lung disease, asthma, emphysema, or chronic bronchitis?" The dichotomous variable presence of one or more mental health conditions (yes/no) was based on the survey questions, "Has a doctor or other health professional ever told you that you had any of the following medical conditions: depression or anxiety disorder?".

#### 2.2.3. Control Variables

The control variables are the sociodemographic characteristics such as gender (Female, Male), race/ethnicity (Non-Hispanic White, Non-Hispanic Black or African American, Hispanic, Non-Hispanic Asian, Non-Hispanic Other), age (18–34 years, 35–49 years, 50–64 years, 65–74 years, 75 years and older), educational level (less than high school, high school graduate, some college, Bachelor's degree, post-baccalaureate degree), household

income (less than USD 20,000, USD 20,000 - < USD 35,000, USD 35,000 - < USD 50,000, USD 50,000 - < USD 75,000, USD 75,000 or more), and rurality of residence (Yes or No).

*2.3. Analysis*

For the categorical variables, descriptive statistics, i.e., frequencies and weighted percentages, were computed to describe the characteristics of the survey respondents. The multivariable Binomial logistic regression was conducted to determine the associations of each of the independent variables—having chronic conditions, mental health conditions, and other covariates in the model with four dependent variables representing social media engagement, after controlling for other variables in the model. STATA version 17 was used for all analyses, and survey weights were applied to account for the complex survey design of HINTS.

**3. Results**

*3.1. Descriptive Results*

Figure 1 depicts the pattern of social media engagement, showing that 70% of the adults used the Internet in the last 12 months to visit a social networking site, such as Facebook or LinkedIn. A much smaller proportion, i.e., 14% used the Internet to share health information on social networking sites in the same time frame. Only 7% of study participants indicated joining an Internet-based support group in the last 12 months for people with similar medical issues. Thirty-five percent used YouTube to watch a health-related video.

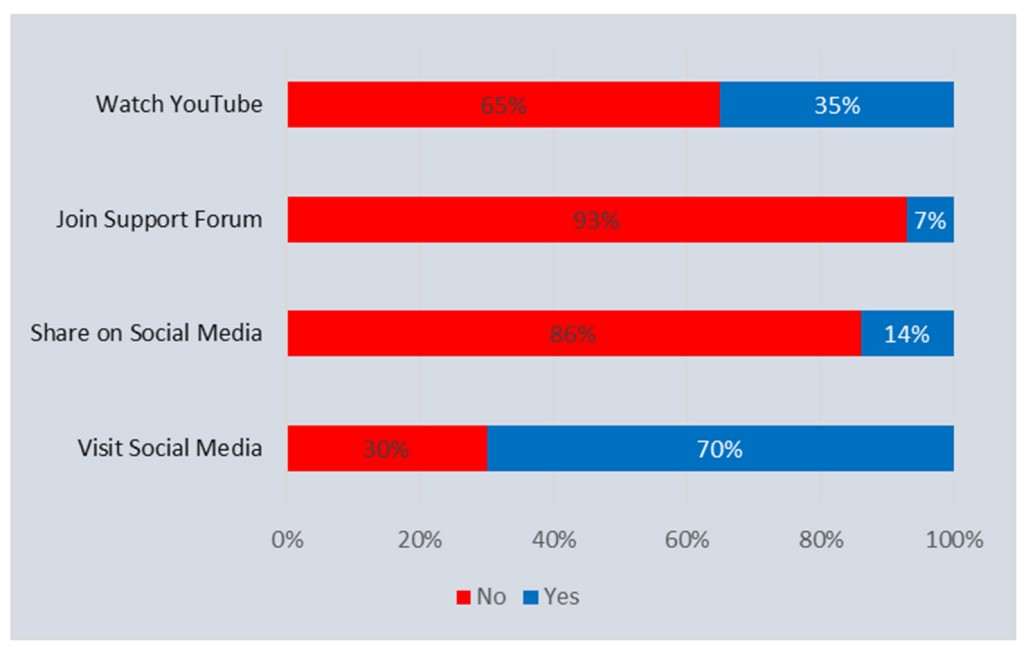

**Figure 1.** Patterns of social media use by adults in the United States, 2017–2020.

In this study population, 48% had at least one chronic condition, and 24% had a mental condition (Table 1). The demographic composition of the study population was 49% men and 51% women. Non-Hispanic White respondents were the largest group (65%), followed by 16% Hispanics and 11% Black. Distribution by age was 18–34 years, constituting 24% of the population, whereas 35–49 years and 50–64 years were, respectively, 27% and 29%. Four in ten (or 39%) of the respondents had an income of USD 75,000 or more, and 87% of the respondents lived in urban areas.

**Table 1.** Descriptive statistics for sociodemographic and clinical characteristics of study participants, 2017–2020.

|  | Frequency (Unweighted) | Percent (Weighted) |
| --- | --- | --- |
| **Chronic Conditions** | | |
| Yes | 8720 | 48% |
| No | 6810 | 52% |
| **Mental Health Conditions** | | |
| Yes | 3585 | 24% |
| No | 12,166 | 76% |
| **Gender** | | |
| Male | 6159 | 49% |
| Female | 8573 | 51% |
| **Race/ethnicity** | | |
| Non-Hispanic White | 9038 | 65% |
| Non-Hispanic Black/African American | 2011 | 11% |
| Hispanic | 2214 | 16% |
| Non-Hispanic Asian | 661 | 5% |
| Non-Hispanic Other | 520 | 3% |
| **Age group** | | |
| 18–34 years | 1944 | 24% |
| 35–49 years | 2984 | 27% |
| 50–64 years | 4986 | 29% |
| 65–74 years | 3452 | 11% |
| 75+ years | 2219 | 8% |
| **Education** | | |
| Less than High School | 1099 | 9% |
| High School Graduate | 2898 | 23% |
| Some College | 4653 | 37% |
| Bachelor's Degree | 4119 | 19% |
| Post-Baccalaureate Degree | 2868 | 12% |
| **Household Income** | | |
| Less than USD 20,000 | 2666 | 17% |
| USD 20,000–USD 34,999 | 1916 | 12% |
| USD 35,000–USD 49,999 | 1880 | 14% |
| USD 50,000–USD 74,999 | 2537 | 18% |
| USD 75,000 or more | 5296 | 39% |
| **Urban residence** | | |
| Yes | 14,135 | 87% |
| No | 1957 | 13% |

*3.2. Logistic Regression of Chronic Conditions and Social Media Engagement*

After controlling for other covariates in the logistic regression model, study participants with at least one chronic condition had significantly higher odds (Adjusted Odds Ratio, AOR, 1.5; CI, 1.11–1.93) of having used the Internet to participate in an online health community or forum for people with similar conditions in the past 12 months compared to those with no chronic conditions (Table 2). People with at least one chronic condition had significantly higher odds (AOR, 1.2; CI, 1.01–1.36) of having watched health-related videos on YouTube in the past 12 months. Having chronic conditions did not significantly change the odds of having used the Internet to visit a social networking site, such as Facebook or LinkedIn. Having chronic conditions also was not associated with having shared health information on social media in the past 12 months.

**Table 2.** Logistic regression of chronic conditions and social media engagement, pooled 2027–2020 Health Information National Trends Survey (HINTS) 5.

| | Visit Social Media | | Share Health Information on Social Media | | Join Support Group for People with Similar Conditions | | Watch a Health-Related Video on YouTube | |
|---|---|---|---|---|---|---|---|---|
| | AOR | CI | AOR | CI | AOR | CI | AOR | CI |
| **Presence of Chronic Diseases** | 1.1 | 0.91–1.35 | 1.2 | 0.99–1.53 | **1.5** | 1.11–1.93 | **1.2** | 1.01–1.36 |
| **Female** | **1.6** | 1.33–1.91 | **2.0** | **1.55–2.59** | **2.2** | **1.58–3.02** | 1.1 | 0.9–1.26 |
| **Race (Ref. Category = Non-Hispanic White)** | | | | | | | | |
| Non-Hispanic Black/African American | **0.67** | 0.52–0.87 | 1.0 | 0.76–1.45 | 1.3 | 0.81–1.99 | 1.2 | 0.97–1.53 |
| Hispanic | **0.62** | 0.47–0.82 | 1.1 | 0.83–1.47 | 0.7 | 0.48–1.12 | **1.6** | 1.26–2.10 |
| Non-Hispanic Asian | **0.64** | 0.44–0.95 | 1.1 | 0.68–1.68 | 0.7 | 0.31–1.38 | **1.7** | 1.12–2.60 |
| Non-Hispanic Other | 0.83 | 0.50–1.39 | 0.8 | 0.49–1.38 | 1.2 | 0.59–2.27 | 1.3 | 0.89–1.87 |
| **Age Group (Ref. Category = 18–34 yrs)** | | | | | | | | |
| 35–49 yrs | **0.51** | 0.35–0.74 | 1.09 | 0.81–1.48 | 1.1 | 0.79–1.59 | **0.75** | 0.58–0.97 |
| 50–64 yrs | **0.29** | 0.20–0.43 | **0.6** | 0.43–0.83 | 0.7 | 0.45–0.96 | **0.54** | 0.42–0.68 |
| 65–74 yrs | **0.13** | 0.09–0.19 | **0.33** | 0.23–0.48 | **0.3** | 0.19–0.58 | **0.33** | 0.25–0.42 |
| 75+ years | **0.05** | 0.04–0.08 | **0.14** | 0.08–0.28 | **0.2** | 0.06–0.40 | **0.14** | 0.10–0.20 |
| **Educational Level (Ref. Category = Less than High School)** | | | | | | | | |
| High School Graduate | 1.2 | 0.77–1.94 | 1.3 | 0.73–2.14 | 1.1 | 0.44–2.62 | 1.2 | 0.80–1.74 |
| Some College | **1.9** | 1.25–2.87 | 1.7 | 1.01–2.87 | 1.7 | 0.77–3.92 | **1.98** | 1.37–2.87 |
| Bachelor's Degree | **2.4** | 1.60–3.47 | 1.7 | 0.99–2.77 | 2.1 | 0.92–4.87 | **2.1** | 1.4–3.0 |
| Post-Baccalaureate Degree | **1.9** | 1.16–2.96 | 1.7 | 0.99–2.78 | **2.8** | 1.19–6.49 | **2.2** | 1.4–3.5 |
| **Household Income (Ref. Category = Less than $20,000)** | | | | | | | | |
| $20,000–<$35,000 | **1.6** | 1.13–2.16 | 1.1 | 0.71–1.61 | 1.03 | 0.60–1.77 | 1 | 0.75–1.33 |
| $35,000–<$50,000 | **1.8** | 1.26–2.55 | 0.9 | 0.55–1.43 | 0.95 | 0.49–1.83 | 0.91 | 0.64–1.29 |
| $50,000–<$75,000 | **1.7** | 1.28–2.21 | 0.7 | 0.47–1.02 | 1 | 0.62–1.61 | 0.9 | 0.66–1.21 |
| >$75,000 | **2.1** | 1.59–2.73 | 0.8 | 0.50–1.29 | 1.04 | 0.69–1.58 | 0.98 | 0.72–1.34 |
| **Metropolitan Area** | **1.3** | 1.03–1.74 | 1.3 | 1.01–1.78 | **1.7** | 1.20–2.53 | 1.1 | 0.86–1.51 |

AOR: Adjusted Odds Ratio; CI: Confidence Interval; AORs in bold font indicate significance at $p \leq 0.05$.

The logistic regression analysis examining the association of covariates in the model also depicts several significant associations with social media engagement (Table 2). After controlling for other variables in the model, women had higher odds compared to men (AOR, 1.6; CI, 1.33–1.91) of having used in the past 12 months the Internet to visit a social networking site, such as Facebook or LinkedIn. Women also had 2 times higher odds (AOR, 2.0; CI, 1.55–2.59) of sharing health information on social media and even higher odds (AOR, 2.2; CI, 1.58–3.02) of joining a support group for people with similar conditions.

The results from multivariable logistic regression analyses also revealed disparities in the use of social media when comparing Whites with minority race groups. Compared to Non-Hispanic White, Hispanics had approximately 0.62 times lower odds (AOR, 0.62; CI, 0.47–0.82) of visiting social media and approximately 2 times higher odds (AOR, 1.6; CI, 1.26–2.10) of watching health-related videos on YouTube; Non-Hispanic Asians had about 0.64 times lower odds (AOR, 0.64; CI, 0.44–0.95) of visiting social media and approximately 2 times higher odds (AOR, 1.7; CI, 1.12–2.10) of watching health-related videos on YouTube; and Non-Hispanic Black/African American had approximately 0.67 times lower odds (AOR, 0.67; CI, 0.52–0.87) of visiting social media.

Compared to the 18–34-year-old age group, respondents within the 35–49-year-old age group had approximately 0.5 times lower odds (AOR, 0.51; CI, 0.35–0.74) of visiting social media, and about 0.75 lower odds (AOR, 0.75; CI, 0.58–0.97) of watching health-related videos on YouTube. Respondents within the 50–64-year-old age group had approximately 0.3 times lower odds (AOR, 0.29; CI, 0.20–0.43) of visiting social media, about 0.6 times lower odds (AOR, 0.6; CI, 0.43–0.83) of sharing health information on social media, and 0.75 lower odds (AOR, 0.75; CI, 0.58–0.97) of watching health-related videos on YouTube

when compared to the 18–34-year-old age group. Compared to the 18–34-year-old age group, respondents within the 65–74-year-old age group had approximately 0.1 times lower odds (AOR, 0.13; CI, 0.09–0.19) of visiting social media, about 0.3 times lower odds (AOR, 0.33; CI, 0.23–0.48) of sharing health information on social media, 0.3 times lower odds (AOR, 0.3; CI, 0.19–0.58) of joining a support group for people with similar conditions, and 0.2 lower odds (AOR, 0.22; CI, 0.25–0.42) of watching health-related videos on YouTube. Respondents above 75+ years had approximately 0.1 times lower odds (AOR, 0.05; CI, 0.04–0.08) of visiting social media, about 0.1-time lower odds (AOR, 0.14; CI, 0.08–0.28) of sharing health information on social media, 0.2 times lower odds (AOR, 0.2; CI, 0.06–0.40) of joining a support group for people with similar conditions, and about 0.1-time lower odds (AOR, 0.14; CI, 0.10–0.20) of watching health-related videos on YouTube compared to those of 18–34 years.

Compared to respondents with less than high school education, respondents with some degree had approximately 2 times higher odds (AOR, 1.9; CI, 1.25–2.87) of visiting social media and about 2-time higher odds (AOR, 1.98; CI, 1.37–2.87) of watching health-related videos on YouTube; respondents with bachelor degree had approximately 2 times higher odds (AOR, 2.4; CI, 1.60–3.47) of visiting social media and about 2-time higher odds (AOR, 2.1; CI, 1.4–3.0) of watching health-related video on YouTube; respondents with post-baccalaureate had approximately 2 times higher odds (AOR, 1.9; CI, 1.16–2.96) of visiting social media, about 3-time higher odds (AOR, 2.8; CI, 1.19–6.49) of joining a support group for people with a similar condition, and about 2-time higher odds (AOR, 2.2; CI, 1.4–3.5) of watching health-related videos on YouTube.

Compared to household incomes of less than USD 20,000, respondents with a household income of USD 20,000-USD 35,000 (AOR, 1.6; CI, 1.13–2.16), USD 35,000 - < USD 50,000 (AOR, 1.8; CI, 1.26–2.55), USD 50,000 - < USD 75,000 (AOR, 1.7; CI, 1.59–2.73) and >USD 75,00 (AOR, 2.1; CI, 1.59–273) had approximately two times higher odds of visiting social media.

Lastly, respondents who lived in the metropolitan area had 1.3 times higher odds (AOR, 1.3; CI, 1.03–1.74) to visit social media and about 2 times higher odds (AOR, 1.7; CI, 1.20–2.53) to join a support group for people with a similar condition.

### 3.3. Logistic Regression of Mental Health Conditions and Social Media Engagement

After controlling for other covariates in the logistic regression model, study participants with depression or anxiety disorder had significantly lower odds of visiting social media (AOR, 0.77; CI, 0.64–0.93) and sharing health information on social media (AOR, 0.75; CI, 0.60–0.94) in the past 12 months compared to those with no depression or anxiety disorder (Table 3). People with depression or anxiety disorder also had significantly lower odds of joining a support group for people with similar conditions (AOR, 0.51; CI, 0.39–0.67) and watching health-related videos (AOR, 0.67; CI, 0.56–0.80) on YouTube in the past 12 months compared to those with no depression or anxiety disorder.

**Table 3.** Logistic regression of mental health conditions and social media engagement, pooled 2027–2020 Health Information National Trends Survey (HINTS) 5.

| | Visit Social Media | | Share Health Information on Social Media | | Join Support Group for People with Similar Conditions | | Watch a Health-Related Video on YouTube | |
|---|---|---|---|---|---|---|---|---|
| | **AOR** | **CI** | **AOR** | **CI** | **AOR** | **CI** | **AOR** | **CI** |
| **Depression or Anxiety Disorder** | **0.77** | 0.64–0.93 | **0.75** | 0.60–0.94 | **0.51** | 0.39–0.67 | **0.67** | 0.56–0.80 |
| **Female** | **1.54** | 1.28–1.84 | **1.89** | 1.49–2.41 | **1.96** | 1.41–2.74 | 1.02 | 0.87–1.21 |
| **Race (Ref. Category = Non-Hispanic White)** | | | | | | | | |
| Non-Hispanic Black/African American | **0.71** | 0.54–0.92 | 1.08 | 0.78–1.50 | 1.42 | 0.91–2.23 | **1.3** | 1.03–1.65 |
| Hispanic | **0.64** | 0.48–0.84 | 1.13 | 0.86–1.48 | 0.8 | 0.52–1.20 | **1.7** | 1.33–2.19 |
| Non-Hispanic Asian | **0.67** | 0.45–0.98 | 1.14 | 0.72–1.79 | 0.78 | 0.38–1.61 | **1.83** | 1.20–2.80 |
| Non-Hispanic Other | 0.83 | 0.50–1.38 | 0.82 | 0.49–1.36 | 1.16 | 0.59–2.30 | 1.27 | 0.71–1.84 |

**Table 3.** *Cont.*

| | Visit Social Media | | Share Health Information on Social Media | | Join Support Group for People with Similar Conditions | | Watch a Health-Related Video on YouTube | |
|---|---|---|---|---|---|---|---|---|
| | **AOR** | **CI** | **AOR** | **CI** | **AOR** | **CI** | **AOR** | **CI** |
| **Age Group (Ref. Category = 18–34 yrs)** | | | | | | | | |
| 35–49 yrs | **0.52** | 0.36–0.72 | 1.14 | 0.83–1.56 | 1.18 | 0.83–1.68 | **0.75** | 0.58–0.97 |
| 50–64 yrs | **0.31** | 0.21–0.45 | **0.64** | 0.45–0.91 | 0.76 | 0.52–1.10 | **0.57** | 0.45–0.72 |
| 65–74 yrs | **0.14** | 0.10–0.20 | **0.37** | 0.25–0.56 | **0.44** | 0.27–0.74 | **0.36** | 0.28–0.47 |
| 75+ years | **0.06** | 0.04–0.09 | **0.19** | 0.10–0.34 | **0.22** | 0.09–0.53 | **0.17** | 0.12–0.23 |
| **Educational Level (Ref. Category = Less than High School)** | | | | | | | | |
| High School Graduate | 1.24 | 0.78–1.96 | 1.3 | 0.76–2.23 | 1.15 | 0.47–2.81 | 1.23 | 0.82–1.82 |
| Some College | **1.92** | 1.26–2.91 | 1.69 | 1.00–2.83 | 1.78 | 0.79–4.00 | **2.04** | 1.41–2.95 |
| Bachelor's Degree | **2.37** | 1.61–3.49 | 1.63 | 0.98–2.71 | 2.19 | 0.95–5.03 | **2.12** | 1.46–3.07 |
| Post-Baccalaureate Degree | **1.85** | 1.17–2.95 | 1.62 | 0.97–2.73 | **2.87** | 1.24–6.66 | **2.28** | 1.46–3.54 |
| **Household Income (Ref. Category = Less than $20,000)** | | | | | | | | |
| $20,000–<$35,000 | **1.6** | 1.16–2.20 | 1.11 | 0.75–1.63 | 1.08 | 0.62–1.88 | 1.07 | 0.80–1.42 |
| $35,000–<$50,000 | **1.86** | 1.31–2.65 | **0.92** | 0.58–1.45 | 1.01 | 0.53–1.93 | 0.95 | 0.66–1.35 |
| $50,000–<$75,000 | **1.78** | 1.35–2.33 | **0.76** | 0.53–1.10 | 1.12 | 0.69–1.80 | 0.99 | 0.72–1.35 |
| >$75,000 | **2.21** | 1.68–2.92 | **0.86** | 0.55–1.36 | 1.17 | 0.79–1.76 | 1.07 | 0.79–1.46 |
| **Metropolitan Area** | **1.32** | 1.01–1.72 | 1.23 | 0.93–1.62 | **1.7** | 1.16–2.47 | 1.11 | 0.84–1.47 |

AOR: Adjusted Odds Ratio; CI: Confidence Interval; AORs in bold font indicate significance at $p \leq 0.05$.

## 4. Discussion

Health information exchange has become the hallmark of modern healthcare systems, and social media is a critical vehicle for such an exchange [27,28]. The exchange of health information for patient self-management is increasingly recognized for its role in improving health outcomes [21,29]. To fill the gaps in the existing literature about the variation in social media use by patients with either chronic conditions or mental health conditions, this study examined social-media-based health-information-seeking patterns such as visiting social media health information sharing, joining online health communities, and watching health-related videos on YouTube. The results of this study showed that individuals diagnosed with chronic conditions were more likely to use the internet for health-related reasons compared to individuals with mental health conditions.

Overall, 70% of adults used the internet to visit a social networking site, such as Facebook, Twitter, or LinkedIn in the 12 months before each cycle of the survey. The results showed that the frequency of social media use by adults is consistent with several other studies [30,31]. A much smaller proportion (14%) used the internet to share health information on social networking sites, and this supports the finding of a nationally representative study of U.S. adults that people have reservations about sharing health information electronically unless their privacy protection and confidentiality are assured, which social media cannot completely assure at all times since there can be security breaches [32]. The use of YouTube for watching health-related videos in less than 50% of the population points to the existence of opportunities for stakeholders to consider. A systematic review of the literature by Madathil and Rivera-Rodriguez shows that people may have difficulty understanding the jargon used in the content or tags of health-related videos on YouTube, and a large proportion of videos may provide scientifically unsound information [33]. The study results showed YouTube as having the highest form of social media engagement after visiting social media, and this is important for public health practitioners and health systems who may need to leverage this platform to disseminate quality health information to the population.

Although 48% of the study participants had one or more chronic conditions and 24% had mental health disorders, only 7% of the study participants had joined online support groups for people with similar conditions, which is perhaps due to a lack of knowledge

about the benefits of participating in such groups and uncertainty about the quality of information available through support groups [34]. While more respondents appear to be more comfortable sharing information on social media, fewer respondents joined support groups for their medical condition; health organizations can explore how feasible it is for them to create and moderate these support groups to help the members improve self-management and increase compliance with their health provider's recommendations.

Our research showed that those with depression or anxiety disorder had lower odds of visiting or sharing health information on social media platforms than those without these conditions. This study finding is in contrast with the results of a study by Alhusseini et al., which reported that people with mental health conditions were more likely to join an online support group [35]. When it comes to mental health, the influence of psychological obstacles in seeking assistance can be barriers to social media engagement, including joining support groups or watching a health-related video on YouTube. The negative stigma and perceptions surrounding mental issues, undermining the seriousness of the problem, trying to be self-reliant, and having difficulty expressing concerns and accessing help can prevent people from social media engagement [36].

Although a past study found that some people with chronic conditions may perceive the information on social media to be spurious and not as reliable as that coming from a healthcare professional and may view online communications as unreliable [37,38], this study showed that respondents who had chronic conditions were more likely to join a support group for people with similar conditions and watch a health-related video on YouTube. This could be due to increased awareness of chronic conditions and more people being interested in health-promoting activities.

In summary, regarding the study objective, the results showed a difference in the pattern of social media engagement, with people with chronic conditions more likely to use social media for health-related reasons than those with mental health conditions.

Regarding ethnic differences in the use of social media, we found that while non-Hispanic Blacks, Hispanics, and Non-Hispanic Asians were less likely to visit social media, Hispanics and Non-Hispanic Asians were more likely to watch health-related YouTube videos when compared to Non-Hispanic Whites. This study finding is consistent with a Public Religion Research Institute report stating that Hispanic adults engage in social media more than Whites [39]. However, findings from other studies are in contrast with our findings; for instance, studies have shown that Whites are more likely to exchange health information using social media sites than other racial groups and are more likely to use social media to seek out health information from others [40,41]. Our study identified that non-Hispanic Asians use social media less frequently than other ethnic groups. These findings suggest that racial disparities exist in the use of social media for health information exchange, potentially caused by socioeconomic inequities such as educational level [42]. A study found that White Americans with college degrees used social media more than those without a college education [39]; addressing the barriers to obtaining education among this racial group may improve engagement. The misinformation promulgated among non-Hispanic Asians due to the dearth of culturally and linguistically appropriate messages on social media may also be a driving factor for the lower rates of engagement [43]. Public health stakeholders should address the underlying causes of racial disparities in the use of social media for health-related reasons. Stakeholders can explore how culturally sensitive health-related programs can be developed and made linguistically appropriate for diverse races on YouTube.

Considering gender differences, we found that, similar to previous studies, women had higher odds of social media use [35,39,40]. The higher odds of social media use by women may be due to the more expressive nature of women [44]. Women were also more likely to share health information on social media and join online health communities since they were more receptive to information posted online [40]. Since women appear to use social media more than men, health-promoting communications can be targeted at women with the hope that they obtain this information for other community members.

Among the sociodemographic factors, we found people older than 18 to 34 years and those with at least some college degree and higher incomes used social media more for health-related reasons. This finding is in line with Kruska and Maresova's study that reported as age increases, the use of social media decreases, and as education and income increase, social media presence also increases [41]. According to the Pew Research Center, young adults are active users of social media platforms [40], and there is a need to make social media user-friendly and engaging for older adults. We found that educated people and high-income earners were more likely to be more interested in the use of social media for health-related reasons. This finding is corroborated by Kruska and Maresova, who reported that rich people engage more in social media networking due to free time availability and for work purposes [41]. Addressing the underlying social determinants of health through policy making that affect social media engagement such as education and household income can lead to more people using social media for health-related reasons.

A major strength of this study is that it provides new information on how an individual's use of social media for health-related reasons may differ based on their health conditions. In addition, the large sample size of the nationally representative study increases the reliability of this study. One of this study's limitations is the secondary nature of the data which limits the variables that can be explored, and another is the possibility that some respondents may not have completely understood the survey questions. Also, because the health conditions of interest and patterns of social media engagement are not mutually exclusive, there may be respondents who have both chronic conditions and mental health conditions and participate in more than one of the social media activities.

## 5. Conclusions

Although chronic conditions and mental health conditions can be debilitating for any individual, identifying as having a mental health condition may lead to stigmatization by other members of the population. Increasing the knowledge of healthcare providers and public health stakeholders on how differently their patients use the internet for health-related reasons may help to understand how social media can be used to deliver healthcare and public health interventions. While the study findings can be applied to improve online patient support groups and public health campaigns, more studies need to be conducted on the facilitators and barriers encountered by patients based on their medical conditions. Also, although some of the survey responses were collected during the pandemic, there is a need to consider if the pattern of social media engagement has changed post-pandemic. Policies that address health disparities, especially those due to the social determinants of health, may lead to increased social media engagement for health-related reasons.

**Author Contributions:** Conceptualization, E.A., G.S., K.K. and B.S.; methodology, E.A. and G.S.; formal analysis, E.A.; data curation, E.A., G.S., K.K. and B.S.; writing—original draft preparation, E.A., G.S., K.K., B.S. and P.S.; writing—review and editing, E.A., G.S., K.K., B.S. and P.S.; supervision, G.S. All authors have read and agreed to the published version of the manuscript.

**Funding:** This research received no external funding.

**Institutional Review Board Statement:** This study was conducted by the Declaration of Helsinki and approved by the Institutional Review Board of Georgia Southern University protocol code H22399.

**Informed Consent Statement:** Patient consent was waived due to the use of secondary data.

**Data Availability Statement:** Research data are available upon request from the authors and publicly available at https://hints.cancer.gov/data/survey-instruments.aspx#H5C4 (accessed on 5 January 2024).

**Conflicts of Interest:** The authors declare no conflicts of interest.

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
