# Peer review of "Variations in Pattern of Social Media Engagement between Individuals with Chronic Conditions and Mental Health Conditions"

_informatics, doi:10.3390/informatics11020018_

Round 1

Reviewer 1 Report (Previous Reviewer 3)

Comments and Suggestions for Authors

The study refers to a pre-Covid-19 world and should have had considered the online turn during the pandemic and how it has affected communication and collaboration about chronic disease and health literacy. 

Author Response

Thank you very much for the comment. Although the data collection included responses from during the pandemic, the authors have provided more information on how the pattern of communication and social media engagement might have changed post-pandemic (Lines 447-449)

Reviewer 2 Report (Previous Reviewer 2)

Comments and Suggestions for Authors

Dear Authors,

Please explicitly indicate in your replies how you have addressed each of the Reviewer comments, including page and line numbers in which you have made each change. Just stating 'Thank you... we have made the necessary revisions' is not a helpful reply at all.

In addition, make sure that what you state in your reply matches to what has actually been done in the manuscript, at the moment there are several discrepancies, please check carefully.

Replies like 'The dataset owners try to ensure that they recruit new participants for every cycle' are not convincing, if anything this needs to be stated explicitly in the text. 

Happy to re-review once these changes are made.  

Comments on the Quality of English Language

A few mistakes still present, please double check throughout. 

Round 2

Reviewer 2 Report (Previous Reviewer 2)

Comments and Suggestions for Authors

None. 

This manuscript is a resubmission of an earlier submission. The following is a list of the peer review reports and author responses from that submission.

Round 1

Reviewer 1 Report

Comments and Suggestions for Authors

This paper examined factors contributing to social media engagement to exchange health information, which is an interesting question. However, the study mainly replicate what have been done by existing studies, and the readers are hard to find meanings beyond empirical data. It was unclear what the study was contributing from a theoretical perspective to health communication or health promotion.

The diagnosis of the gaps in the existing research is mostly empirical while the whole study mainly replicated what have been found in the previous studies. The authors should further explain the creativity of the current study.

The selection of variables and measurement design needs further explanation. For instance, how were the four variables measuring social media engagement selected? Why was “the presence of mental health” measured only by two medical conditions, i.e. depression and anxiety disorder? 

The findings and discussion are mainly descriptive. After reading the whole paper, the readers are hard to find meanings or theoretic contributions beyond empirical data. The author must add a new chapter for in-depth scholarly discussions. 

Special difference of online health information exchange between this paper and other researches should be discussed in the current study.

Reviewer 2 Report

Comments and Suggestions for Authors

Many thanks for giving me the opportunity to review this manuscript setting out to associate demographic characteristics and presence of physical and/or mental, long-term conditions with social media engagement for health-related purposes. The topic is of importance because, as authors also note, patients are increasingly using social media to seek health-related information and support from other peers, a fact that might well impact on the structure and the delivery of healthcare. Therefore, the topic might be of interest for the readership of the ‘Informatics’ Journal.

Unfortunately, several parts of the manuscript are poorly structured and important pieces of information are missing, e.g., in relation to how recruitment and data collection were conducted. In addition, it is not clear for what type of conditions this study has accounted. By reading the Introduction it seems that every single long-term condition was of interest for this study, by reading section 2.1. it seems that only cancer was of interest, and then by reading section 2.2.2. you start realising that certain long-term conditions were of interest. Have you done any analysis to explore variation based on the condition, e.g., is social media engagement different in different conditions (e.g., in cardiology-related problems vs asthma etc.)?     

It is unclear what gaps in knowledge authors are trying to address as well as where do findings of this study stand in the greater picture (in terms of literature on the topic) and how they could be employed and in which settings, and for what purposes. In other words, what audiences are your findings targeting? All these matters need to be thoroughly explained in the Discussion section, which needs to be re-structured and more explicitly discuss the findings of the study. Currently, the Discussion section consists of several arguments, but these are somehow lost, and/or they are often unclear.  

In any case, I hope that my comments below are of some help in improving the quality of the manuscript and look forward to seeing this manuscript published.     

Title:

-The way title is framed implies that ‘mental health’ is NOT a ‘chronic disease’. I do understand the differentiation from an analysis point of view, but sounds bizarre to the readers. Can you re-phrase slightly? E.g., ‘… by chronic disease, mental health and socioeconomic status, and …’.

-‘Minority race’ is such an odd term. Belonging to a certain race does not necessarily make you a ‘minority’. Suggest you remove ‘minority’ and just keep ‘race’.

-It is usually good to indicate the study design at the Title level. Might be worth re-phrasing the title to something like ‘Effect of … on health information exchange via social media: (indicate design)’.

Abstract section:

-Line 12: ‘exchange of health information searches’, this is very unclear. Either remove ‘searches’ or explain the concept.

-Line 12: ‘accordingly’ to what? It is unclear. You can possibly get rid of the word ‘accordingly’.

-Line 13 and throughout: In healthcare, it is usually good to refer to ‘patients’ rather than ‘consumers’. We are, or we should try to be, in a patient-centric era! I suggest you remove the term ‘consumers’ throughout.

-Lines 14 and 15: As per my previous comments in relation to ‘mental health’ being amongst ‘chronic health conditions’, I suggest you re-phrase to something like ‘… individuals diagnosed with chronic diseases, including mental health conditions’.

-Line 15: You will need to explain what does ‘secondary data’ mean, this is totally unclear.

-Line 16: You will also need to explain where does this numerical value refer to? Readers will be unclear as to whether this figure refers to respondents, clinical records, aggregated data etc.

-Line 17: I suggest you state, in brackets, the independent variables you are talking about.

-Lines 23-24: I suggest you re-phrase this sentence to ‘Race, age, and educational level also influence …’ and if you have the space, maybe slightly expand to explain in what terms.

-Lines 24-26: The final sentence is very unclear and really difficult to follow, e.g., what does ‘are targeted’ mean, ‘targeted’ to what?; ‘engagement’ with what?; what does ‘engagement pattern’ have to do with ‘public health interventions’? Please re-phrase this sentence, makes no sense at the moment.

-Lines 25-26 and throughout the manuscript: ‘certain health conditions’ – what were those? Have you focused specifically on certain long-term conditions? If so, what were these? You will need to be specific, just stating ‘chronic disease’ means very little. There are countless long-term conditions.    

Introduction section:

-Line 32: ‘for a year or longer’, this sounds really arbitrary. Chronic conditions might influence daily activities life-long. Suggest you re-phrase and/or expand. In addition, explain the type of ‘activity’ you are referring to with ‘limited activity’.

-Line 35: ‘To illustrate’, what are you trying to ‘illustrate’? Unclear at the moment.

-Lines 36-38: This sentence is confusing in the sense that ‘co-morbidities’ might be ‘chronic conditions’ themselves. Please consider re-phrasing as it does not make sense.

-I suggest you keep terminology consistent throughout as it is getting confusing for the readers: ‘chronic disease’, vs ‘chronic health condition’, vs ‘chronic condition’, vs ‘chronic medical condition’. Globally, ‘long-term condition(s)’ is the most widely used term. Likewise, keep it consistent how you refer to mental problems: ‘mental health’, vs ‘mental health conditions’ vs ‘mental conditions’, vs ‘mental health illness’, vs ‘mental illness’, vs ‘behavioural illness’, vs ‘mental health disorders’.    

-Line 38: ‘These chronic diseases’, where does ‘these’ refer to? Please explain or re-phrase.

-Line 39: ‘spending’ is a very vague term. Do you mean ‘healthcare costs’?

-Lines 40 and 41: Once more, please consider re-phrasing this sentence as at the moment it seems like ‘mental health’ is not a ‘chronic condition’. Mental health problems may or may not be chronic. Perhaps somewhere at the end of the Introduction or in the Methods section explain that for the purposes of this study, you will be treating ‘mental health’ separately from other ‘long-term conditions’. Alternatively, and maybe more straightforward to the readers, you may well say ‘long-term physical conditions’ vs ‘mental health conditions’.

-Lines 42-43: Not sure what this sentence is trying to convey. The term ‘co-morbidities’, by definition, means ‘conditions existing together’. What are the authors trying to say here? I am confused.

-Line 48: Remove the word ‘These’.

-Lines 48-50: I suggest you briefly define these syndromes/conditions (e.g., within brackets) as readers may have no clue. Have you accounted for all of these in your study? Need to be specific, just stating ‘mental health’ throughout the manuscript means very little, i.e., what sort of mental conditions have you taken into consideration?     

-Lines 51-54: The authors state that the prevalence of mental health problems has worsened, and they quote some figures. However, just quoting some figures does not necessarily indicate if something has worsened or improved, e.g., depression prevalence at 31.4%, this does not point out any improvement or worsening. You will need to show the change. In addition, which ‘meta-analysis’ are you talking about? Will need to explain.

-Line 55: where does ‘highest’ and ‘lowest’ refer to? Do you mean prevalence of mental health problems? Please consider re-phrasing.

-Line 62: ‘Social media is used’, ‘used’ by whom?

-Lines 64-66: I am not sure what does the sentence beginning with ‘health information seeking’ add. To me, it is just a repetition of the previous sentence and complicates the flow.

-Lines 66-70: what does ‘clinical care of others’ (line 68) mean? I am totally unclear. In addition, the word ‘including’ is usually used when provision of examples is desired. Are the examples you are providing (lines 68-70) referring to self-care or to care of others? In general, this is a very complicated and unclear sentence. You will also need to define ‘participative or consumer-oriented health care’, ‘cost-containment’, and ‘comparative “shopping” of care’, these are all very unclear terms.

-Line 78: what sort of ‘people’ you are talking about?  

-Lines 80-82, please re-structure this sentence so to better link with previous argument about shared decision-making. In fact, these are the two sides on the coin, i.e., access to online information makes patients better prepared for contact with clinicians but, on the other hand, patients might no longer view clinicians as ‘experts’.

-Lines 82-84: This argument about clinicians needing to deal with increased volume of questions by patients is very important but it somehow gets lost the way it is being currently presented. Can you slightly re-phrase so to better relate it to the clinician-patient relationship?  

-Line 86: Remove ‘their’ before ‘prevention’.

-Line 87: Remove ‘by searching the internet’ as this is obvious in the sentence.

-Lines 87-88: Is this sentence referring to communication of patients with providers and other patients? Please make it clear, and as per my previous comments in relation to the term ‘consumers’.

-Line 88: Where does ‘this information’ relate to? What type of ‘information’ you are talking about?

-Line 89 and throughout: In relation to online health communities, please keep terminology consistent, i.e., ‘online support groups’, vs ‘support groups’, vs ‘online forum’, vs ‘support forum’, vs ‘forum’, vs ‘internet-based support group’, vs ‘chronic-condition specific virtual communities’, vs ‘online communities’, vs ‘disease-specific support group’. I do understand that the relevant survey question referred to ‘online forum or support group for people with a similar health or medical issue’, however, bear in mind that the most frequent terms in the literature, which are also self-explanatory, are ‘online health community’ or ‘online heath forum’.

-You will need to describe what type of online health communities you have accounted for in your study. Were they open or closed groups? Bear in mind that previous literature reports that patients with mental health problems are eager to visit closed communities. If you do not have any information on the type of online health communities, you will need to acknowledge this as a limitation by also emphasising the likelihood of participants not having understood the relevant survey question (i.e., ‘online support group’ does not necessarily mean anything to patients).

-Lines 90-92: In that sentence, authors are mentioning that health information usage depends on ethnicity, gender, age, and socioeconomic status and they then move on by explaining the precise correlation with gender. What about the other variables? It is odd only describing the precise association for just one of the variables.

-Lines 92-94: Please strengthen this argument to emphasise the difference/importance of your study. The novelty of your study is that you examined associations between social media use and demographics in patients with long-term conditions, as compared to the general population (done in previous studies). Is that correct? If so, please make it clearer to emphasise the gap in knowledge you are trying to address.

-Lines 94-97: As a reader, I am struggling to understand the first objective, can you somehow simplify/make it more specific? E.g., ‘assess the patterns of social media engagement for health-related purposes, specifically, visiting social media, sharing health information on social media, joining a support group…, and watching…’.

-Line 99: Remove the word ‘such’.

Materials and Methods section:

-Line 107: So, was ‘cancer’ the only long-term condition you have accounted for? What does ‘use of cancer’ mean?

-Line 108: What does ‘non-institutionalized Americans’ mean? In addition, bear in mind that the term ‘Americans’ implies residents of a continent rather than a country, wasn’t this a US-based study solely?  

-Lines 108-109: Explain what do you mean with ‘two-stage sampling design’, this is totally unclear to the readers.

-Line 109: The way recruitment is presented gives the impression that response rate was 100%. Any idea about the response rates in each year? Some additional details about recruitment will be helpful, e.g. how was the survey distributed? Where did you recruit participants from, i.e., from what settings? These details are currently missing.  

-By reading section 2.1., it seems that there is a chance that same people have participated more than once (i.e., participation in multiple years). Have the authors accounted for this scenario? If this was a possibility indeed, you will need to acknowledge it amongst as a limitation of the study.

-In general, some more detail about the survey structure/items etc. will be helpful for the readers, I believe.  

-Line 112: ‘deemed the study exempt’, ‘exempt’ from what? Please also note that your statement in here about the study being exempt sounds contradictory to your statement on lines 351-353 about the study being approved by the Institutional Review Board of Georgia Southern University. Statement on lines 351-353 need also to be made explicit.    

-Line 149: Suggest you state again the four ‘categorical variables’ you are referring to.

-Line 153: Suggest you re-iterate the ‘four dependant variables’ in here.

-Somewhere, you will need to define the term ‘social media’ and explain why YouTube is, or might be, classified as ‘social media’.  

Results section:

-Line 164: I cannot see how ‘health issues’ are different to ‘medical issues’.

-Line 167: Figure legend is really complicated, please consider simplifying, e.g., ‘Patterns of social media use’.

-The four dependent variables are not necessarily mutually exclusive. E.g., ‘joining a support group’ might well entail ‘sharing health information’. Likewise, there might be overlaps between ‘joining a support group’ and ‘visiting social media’. I do understand that this was how these variables were collected in the survey, but maybe somewhere explain the possibility of overlaps (e.g., in a Limitations section in the Discussion).

-Line 174: Which metropolitan area you are talking about?

-Lines 186-187: Any examples of ‘health information’ shared? Was anything mentioned in the survey?

-Line 203: ‘visit social media’ only, what does ‘only’ mean?

-Check grammar on line 214.

Discussion section:

-Line 254: ‘modern healthcare enterprise’, this is a very vague and complicated term. Please simplify. Healthcare should not be treated, at least in an ideal world, as business.

-Lines 254-255: Please specify what do you mean with ‘health information exchange’, i.e., where and when and in what terms?

-Line 256: ‘clinical decision support tool’, what does this mean? Do you mean for patients or for clinicians? As survey data was collected from patients, I believe that all findings relate to patients and this needs to be made clear throughout.

-Lines 257-258: ‘efficient healthcare coordination’, ‘cost reduction’, ‘quality and safety of healthcare’, all these are non-self-explanatory terms, and you will need to explain.

-Line 259: What are these ‘gaps in literature’ you are talking about?

-Lines 263-264: I am not sure what this sentence (last sentence in first paragraph of the Discussion) means/adds.

-In general, usually the first paragraph of the Discussion summarises the study’s findings. At the moment, authors are just re-iterating the study’s objectives and attempting to ‘defend’ their study, which does not make sense at this stage of the manuscript.

-Line 266: What ‘activity(ies)’ you are talking about? It is totally unclear to the readers.

-Line 268: ‘in the 12 months before the survey’, however, the survey was carried out over multiple years and statements like such are confusing. Please consider re-phrasing.

-Line 268: Which ‘systematic review’ you are referring to? Your study was not a systematic review.

-Line 269: ‘this level of use’, which ‘level of use’ you are referring to? Please consider simplifying the whole sentence, it is very difficult for readers to follow.

-Line 270: What does ‘understandably’ mean? Consider re-phrasing.

-Lines 271-272: ‘This is consistent’, what is ‘consistent’? You haven’t explored patients’ opinions on sharing health information as such. Please consider re-phrasing.   

-Lines 273-274: I am not sure about the statement in relation to social media not assuring privacy, was this indeed mentioned in reference 32? There are many online health communities (on Facebook, Reddit, etc.), which are closed and hence privacy is ensured (to a large degree).

-Lines 274-278: This sentence is really large and complicated, e.g., why 35% is an encouraging percentage? What ‘gap’ you are referring to? Not sure what does ‘fill this gap’ mean. Nobody guarantees that the existence of additional material on YouTube will translate to more people accessing it. How is ‘credibility’ you are referring to determined? Nobody mentioned in the survey that they refrain from watching YouTube educational material because they are not credible, or am I wrong?

-Line 279: What does ‘common people’ mean? Very vague term.

-Lines 285-286: ‘unregulated quality of information available through support groups’, I am not sure about this statement. There are several online health communities which are well-moderated (even by healthcare professionals) and the available information is sound and decent. Many professional bodies and service providers have also published guidelines on accessing and sharing information on social media, both from a patient and a clinician perspective.

-Line 286: Which ‘finding’ you are referring to with ‘this finding’?

-Line 289, reference 35. There are many other studies though that found the opposite, i.e., the increasing use of online health communities amongst patients with long-term conditions. Consider presenting both sides/arguments.

-Line 289: ‘This contrasts with our research’, where does ‘this’ refer to? What does ‘our research’ mean? Do you mean your study or other pieces of research you have done?

-Line 295: Change ‘further’ to ‘furthermore’.

-Lines 295-297: Once more, bear in mind that many online health communities are moderated by healthcare professionals, and this is known to the patients. Therefore, I am not sure about all patients not trusting these communities. There are different findings in this literature, worth exploring and presenting all available findings/arguments.

-Line 305: What findings you are referring to with ‘these findings’? Your findings or findings of other studies? Please make it clear.

-Lines 306-307: Define ‘structural socio-economic inequities’, it is unclear.

-Line 307: Change ‘further’ to ‘furthermore’.

-Line 307: Where does ‘this was found’ refer to? Was it found in your study or the literature?

-Lines 308-309: Please link that sentence in a more appropriate way with the previous one. It is not clear if it is a separate argument or if it follows from the previous one about White people with college degrees.

-Lines 310-311: ‘dearth of culturally and linguistically appropriate messages on social media’, please either explain this section of the sentence and/or simplify the whole sentence.

-Line 314: Check grammar in the following section of the sentence ‘… as in the case with…’.

-Lines 315-316: ‘higher inclination to engage in interaction stem from the naturally more expressive nature of women’, there are several problems with this section of the sentence: 1) ‘interactions’ with whom and where? 2) What does ‘expressive nature’ mean? Do you mean extroversion?

-Lines 318-320: Correct grammar in that sentence.

-Lines 320-322: Correct grammar in that sentence.

-Line 323: What does ‘early adopters’ and ‘active users’ mean? Please explain these terms.

-Lines 324-325: What do ‘proactively seeking information’ and ‘health-related communications’ mean? These are non-self-explanatory terms, please explain.

-Line 326: Once more, please keep terminology consistent, i.e., what does ‘social media channels’ mean?

-In general, I feel that the Discussion section needs to be written in a more concise way. At the moment, there are many arguments but these are often scattered all over the place and/or they do not link with each other, thereby important points are being lost. Authors need to make clear in the Discussion section what are their principal findings, and where do they stand in the greater picture of the topic (i.e., what do they add/how do they compare with existing literature). Authors will also need to discuss ‘Strengths and Limitations’ and ‘Implications’ of their study by creating relevant sections in the Discussion.

Conclusion section:

-Line 331: It is very odd having a reference in the Conclusion section, Conclusion sections stem from study findings, not from the literature.

-It feels like the Conclusion is not grounded to your findings. Conclusion sections need to emphasise the ‘imprint’ of your findings on policy, practice etc. In other words, what your findings have actually added to existing knowledge? At the moment, as a reader, I feel that the Conclusion consists of irrelevant statements. For example, what do your findings have to do with: percentages of long-term conditions in the USA (lines 330-331); presence of misinformation in social media (lines 335-336); ‘positive’ versus ‘deleterious’ information (lines 337-338); and communication amongst healthcare professionals (line 340)?? What is the relevance of your findings to the improvement of e-health literacy? The fact that certain cohorts of people were not found using social media does not automatically translate to limited e-health literacy. Please re-write a more concise Conclusion and base it on your findings.

Comments on the Quality of English Language

Overall, quality of English is acceptable but there are some grammatical mistakes here and there. I have indicated some in my comments, but please check the whole manuscript carefully.  

Reviewer 3 Report

Comments and Suggestions for Authors

The study refers to data representing a pre covid world (2017-2020), while would be much useful to include further and updated data. The dependant variables about social media networks usage and the dicothomy yes/no in the answers make no possible to understand how much and how often participants used social media (daily, weekly, once per month, etc..) given strong ratio of social media daily users (cfr. https://www.pewresearch.org/internet/2021/04/07/social-media-use-in-2021/). Moreover, data about the contents of conversations, videos, posts, etc.. would have provided much sounded picture about the phenomenon (cfr. https://www.ncbi.nlm.nih.gov/pmc/articles/PMC9146886/ and https://www.ncbi.nlm.nih.gov/pmc/articles/PMC8156131/), as the World Wide Web gives access to relevant information based on empirical knowledge, as much as to personal opinions and fake news as stated in the conclusions. The role and engagement of healthcare professionals in the survey is also missing as dimension considering how social media can empower and engage patients to explain better forms of disparities and practices of exclusions enacted by digital means. 
